# Factors shaping subjective financial well-being in emerging adults: A comparative study of Italy and Germany

**Maria Rosa Miccoli**[1]*, **Yury Shevchenko**[1], **Paola Iannello**[2], **Ulf-Dietrich Reips**[1]

**1** Department of Psychology, Research Methods, Assessment, and iScience, University of Konstanz, Konstanz, Germany, **2** Department of Psychology, Università Cattolica del Sacro Cuore, Milan, Italy

* maria-rosa.miccoli@uni-konstanz.de

## Abstract

Academics and policymakers recognize the growing importance of subjective financial well-being for emerging adults, yet little is known about the factors influencing the subjective financial well-being perceived by the emerging adult population. We aim to investigate the role of socio-demographic characteristics (job type, age, country of residence), individual differences in financial knowledge (financial literacy), skills (financial behavior), cognitive response styles (impulsiveness, future orientation, and maximization), and attitudinal components (trust in governmental institutions and financial professionals) in shaping subjective financial well-being among emerging adults in Italy and Germany, representing the Mediterranean and Northern models of the transitions to adulthood, respectively. A sample of 385 participants residing in Italy ($n$ = 193) and Germany ($n$ = 192) voluntarily participated in an online survey. Variables such as trust and maximization were incorporated into a prior financial well-being model to assess their relevance in predicting subjective financial well-being. In Model 1, variables from the prior theoretical model (socio-demographic characteristics, financial literacy, financial behavior, impulsiveness, and future orientation) were analyzed. In Model 2, trust and maximization were added as predictors of subjective financial well-being. Results revealed that the inclusion of these variables improved the model fit, and further confirmed the significant role of age, financial behavior (specifically, caring for financial matters), impulsiveness, future orientation, and trust in governmental institutions in subjective financial well-being.

## Introduction

Emerging adults, that is, individuals aged 18–29 who occupy a transitional stage of life between adolescence and adulthood, represent a unique population requiring specific attention in well-being studies due to the high levels of anxiety and uncertainty that characterize this stage of life. In terms of financial well-being, the transition to adulthood is marked by an overestimation of future financial situation [1] and self-evaluation based on comparison with peers [2]. These factors play a significant role in financial behavior and identity formation, challenging the levels of experienced financial well-being. Recent studies have shown a positive relationship between subjective financial well-being and psychological well-being

**Data availability statement:** The data are held and accessible at the Open Science Framework. Here is the URL to the project: https://osf.io/hmqfe/

**Funding:** The author(s) received no specific funding for this work.

**Competing interests:** The authors have declared that no competing interests exist.

[3]. Understanding the factors that contribute to subjective financial well-being is crucial for enhancing both subjective and psychological well-being in the emerging adult population. Despite the considerable focus on emerging adulthood within the psychology field, there is a gap in understanding the specific factors that are related to financial well-being and whether outcomes differ across various European socio-educational models of transition to adulthood (i.e., different average ages for leaving the parental home, key indicators of perceived adulthood, etc.), such as the Mediterranean and Northern model [4]. In the Northern model, individuals often achieve economic independence at a younger age due to public support or greater access to job opportunities. In addition, there is a long interim period between leaving the parental home and starting a new family. In contrast, the Mediterranean model is characterized by a tendency to transition directly from the parental home to a new household [5].

First, it is essential to define financial well-being and recognize its increasing significance in the modern era, not only for individuals but also for society as a whole. As economic risks have progressively shifted from the state to the individual [6,7], with Europe undergoing substantial pension reforms since 2000, the importance of focusing on financial well-being has grown. Additionally, the Easterlin paradox highlighted a lack of correspondence between rising global prosperity and individual well-being [8,9]. In response, policy-makers, as well as experts in social sciences and psychology, have sought to identify the most effective indicators of financial well-being in the general population.

For a long time, financial well-being has been defined in terms of material resources. However, focusing solely on these objective aspects suggests that material resources are the exclusive indicator of financial well-being. This perspective implies that two emerging adults with identical economic resources would have the same level of financial well-being, regardless of societal or individual differences. Still, research demonstrated that other dimensions of life — such as social interactions and personal safety, which can be influenced by different models of transition to adulthood [4] — also impact overall well-being. Furthermore, society as a factor should be considered given the close relationship between financial well-being and income distribution [10].

The European Commission's "GDP and Beyond" initiative [11] sought to explore how social policy objectives could better foster well-being. Consistent with this approach, one might expect that two emerging adults with identical economic resources within the same society would experience the same level of financial well-being. Yet, this contrasts with extensive literature suggesting that subjective financial well-being, along with individual differences, should also be considered integral components of the broader concept of financial well-being [12]. A first research question arises: How are socio-demographic characteristics (job, age, country of residence) related to financial well-being?

Drawing on the definition of financial well-being as multidimensional, its subjective aspect refers to the perceived level of personal security and freedom of choice, based on individuals' emotional and cognitive evaluation of their financial situation [12]. Kempson and colleagues [13] developed a comprehensive model of financial well-being, which clearly outlines key macro-areas that predict the general construct: socio-demographic characteristics, knowledge and skills (financial literacy and behaviors), but also psychological factors (e.g., time orientation, impulsivity, social status, self-control, locus of control, and attitudes toward spending, saving, and borrowing). By integrating these individual-level factors (psychological factors, skills and financial behaviors), the model highlights the need to view financial well-being comprehensively, acknowledging that individual well-being is deeply interconnected with broader societal influences. This approach offers a broader perspective on financial well-being, positioning individual financial outcomes within the larger societal context. A second research question to explore is whether adopting validated scales measuring financial literacy

and behavior (personal knowledge and skills), alongside psychological factors such as impulsiveness, future orientation, and maximization can confirm their relationship with subjective financial well-being.

While Kempson and colleagues' model [13] provides a solid foundation, it could be further improved by incorporating an element that explicitly connects the individual and society at the institutional level —namely, trust. Trust in governmental institutions and financial professionals shapes people's cognitive evaluation of shared interests in society [14], which is closely tied to perceived safety. For emerging adults, financial independence is a key priority in financial well-being, often viewed as a tangible goal. However, two emerging adults in the same society and with similar individual characteristics (financial literacy and behavior, impulsiveness, time orientation, and maximization) might experience their pursuit of financial independence differently depending on their degree of trust in governmental institutions and financial professionals. Research has shown that trust in governmental institutions [15] and financial professionals [16] is crucial for increased efficiency in navigating complex decisions and enhanced financial well-being. Thus, a relevant third research question that remains unexplored is whether trust in governmental institutions and financial professionals is associated with subjective financial well-being.

The multidimensional nature of financial well-being [12] is well recognized, yet there remains a lack of research on its subjective aspects, highlighting the need to address this gap by focusing on specific predictors of subjective financial well-being that have long been neglected. Previous studies have shown that the same predictors affect objective and subjective financial well-being in different ways [17]. Thus, focusing exclusively on the subjective side of financial well-being may reveal which factors are involved and clarify their specific importance.

This research contributes to the literature in several key ways. Foremost, it investigates whether the two main European models of the transition to adulthood are differently related to subjective financial well-being. While scholars have acknowledged the importance of identifying well-being indicators among this population segment and the potential differences determined by the models of transition to adulthood [4], these differences have not been systematically examined within the context of subjective financial well-being. Second, we compare the indicators from a previous model of financial well-being [13] with an advanced model that employs more specific scales to assess these indicators, aiming to evaluate which are the specific factors that are involved in subjective financial well-being and their relevance. Third, we included trust in governmental institutions and financial professionals into the analysis to assess its relationship with subjective financial well-being. While the role of trust has been previously explored [16], it has not yet been integrated into a model that combines both individual and social indicators of subjective financial well-being.

The overarching aim of this study is to contribute to the development of a comprehensive model of subjective financial well-being model for the emerging adult population, integrating socio-demographic and psychological components with attitudinal factors.

## Theoretical analysis and hypothesis development

Many studies have focused on single factors related to subjective financial well-being, that have often considered each factor separately. To attain an integrated and comprehensive overview of the factors involved in subjective financial well-being, the primary goal of this study is to develop a model that incorporates both socio-demographic and psychological (attitudinal and behavioral) components. A secondary objective is to understand the specificities of subjective financial well-being within the emerging adult population in countries that represent different socio-educational models of transition to adulthood in Europe —namely, the Mediterranean and Northern models [4].

In the following sections, we present an analytical framework grounded in previous research that helps clarify the role of the proposed independent variables in subjective financial well-being. First, we define the concept of subjective financial well-being as employed in this study. Next, we describe the key factors that are relevant to subjective financial well-being in the context of the current study: socio-demographic characteristics (country of residence, age, job type), financial literacy, financial behavior, cognitive response styles (impulsiveness, future orientation, and maximization), and trust (toward governmental institutions, and financial professionals).

## Subjective financial well-being definition and measure

Financial well-being is characterized by multidimensionality, and it comprises both an objective and a subjective dimension [12]. Objective financial well-being refers to the tangible material resources determined by the balance between income, expenses, and accumulated assets. In contrast, subjective financial well-being is defined as an individual's perceived sense of security and freedom of choice, shaped by their emotional and cognitive evaluation of their financial situation. Furthermore, subjective financial well-being is related to individual differences and personal attitudes toward financial conditions [18]. The emotional and cognitive assessment of financial situation changes throughout life stages, for example, during the transition to adulthood. Emerging adulthood, a phase between adolescence and adulthood, is of particular interest in psychological studies due to its transitional nature [12]. Furthermore, this phase is often marked by low levels of subjective and psychological well-being, largely due to increased anxiety and uncertainty.

In this study, we employ the "General Financial Well-Being" measure from the *Multidimensional Subjective Financial Well-Being Scale* for emerging adults [9]. This measure was selected for its recognition of the multidimensional nature of financial well-being and its specific relevance to emerging adulthood. Sorgente and Lanz [9] emphasized that the aforementioned measure can effectively be used on its own to assess subjective financial well-being in a multidimensional framework, given its significant relationship with objective financial well-being.

## Factors of subjective financial well-being: Theoretical and analytical framework

We propose a subjective financial well-being model (see Fig 1), which illustrates the hypothesized factors related to subjective financial well-being in emerging adulthood. By pursuing the goal of investigating the factors that exclusively predict the subjective dimension of financial well-being, we extend the theoretical framework of Kempson and colleagues [13] by delving into the role of specific psychological factors and include trust in governmental institutions and financial professionals in the current work [16,19].

Socio-demographic characteristics are a combination of social and demographic factors of a specific group or population. Age and country of residence contextualize people's economic perceptions and situations [20,21], which are strongly influenced by the social reference environment, especially during transitional phases like emerging adulthood [4]. The job type also affects access to financial resourcesthrough its influence on job satisfaction [22].

Previous research has shown that age [23], country-specific social systems [18] characterizing the countries considered for the study, Italy and Germany, and levels of trust in social and financial institutions vary across countries and may affect subjective financial well-being. For example, Italy reports low trust in banks, while Germany exhibits high trust in financial institutions, including banks and financial professionals [24]. We expect to observe differences in subjective financial well-being based on country of residence, age, and job type.

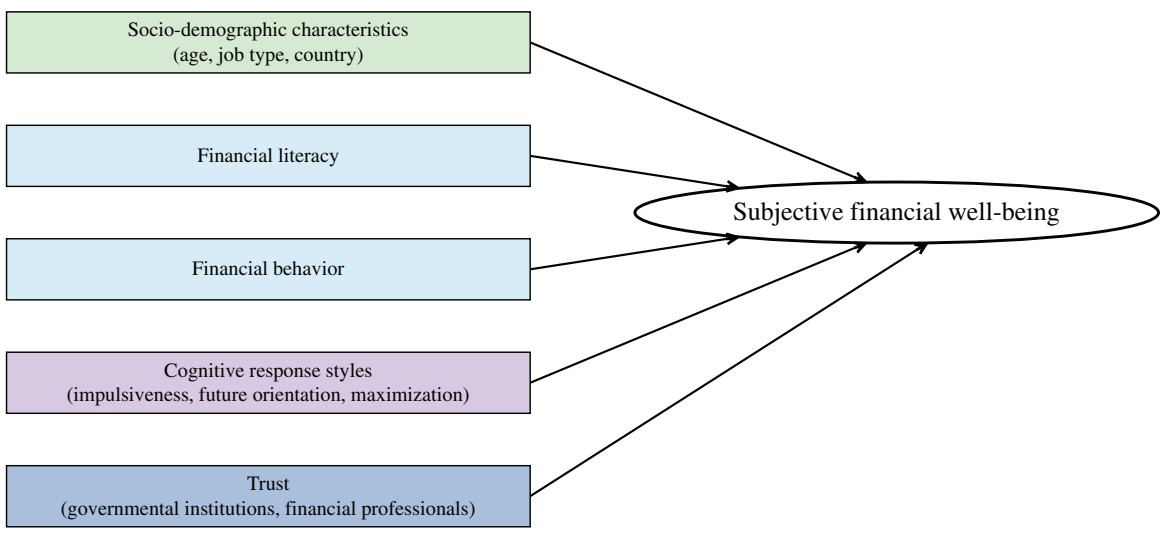

**Fig 1. Factors of Subjective Financial Well-Being in Emerging Adulthood.** *Note.* The proposed model illustrates the hypothesized factors influencing subjective financial well-being. The factors include socio-demographic characteristics, individual components such as financial literacy and behavior (i.e., personal knowledge and skills in the financial domain), individual differences in cognitive response styles, and attitudinal components (trust).

*Hypothesis 1: Socio-demographic characteristics (age, job type, and country of residence) correlate with subjective financial well-being. Differences in subjective financial well-being are hypothesized across countries with significantly different social systems and attitudes towards them. Higher age and job security are hypothesized to positively correlate with high perceived financial security.*

To understand how personal knowledge and skills in the financial domain (i.e., financial competencies) impact subjective financial well-being, we examine financial literacy and behavior.

Financial literacy is defined as the understanding of financial theoretical knowledge and procedures. It is measured by testing the knowledge about concepts as interest, inflation, stock market, time value of money, risk and return, and portfolio diversification. Financial literacy positively influences individuals' cognitive evaluations of their financial situation, an important component of subjective financial well-being [12]. High financial literacy supports informed economic decisions and promotes proactive [25] and virtuous financial behaviors [26,27]. However, experts highlight the need to extend the research to better define reliable measures of financial literacy to incorporate both perceived financial knowledge and objective financial literacy [28,29], due to the inconsistencies between measures of financial literacy and financial behavior [30].

Despite the lack of consistency in results [31], we consider it essential to evaluate the role of financial literacy given the generally demonstrated positive correlation with subjective financial well-being [27] and financial behavior [25,32]. Additionally, examining this relationship is fundamental for emerging adults due to the numerous financial literacy programs offered from early education [33].

*Hypothesis 2: Financial literacy is hypothesized to be positively related to subjective financial well-being.*

Financial behavior refers to individual's actions within financial contexts, such as the intentional use of financial products [34], decisions made over time (e.g., earning, saving, spending,

investing) and in relation to specific events, like retirement [35,36]. Thus, financial behavior was chosen to investigate the role of skills and engagement in financial activities for subjective financial well-being. It is measured through questions investigating how people behave in the financial domain, regarding purchases, bills, budgets, but also savings, and loans. Implementing target behaviors in a specific domain enhances personal security in the corresponding domain [37,38]. Engaging in effective financial behaviors from a young age positively impacts financial well-being [39] and enhances personal security [40].

Previous research has shown a strong link between domain-specific behavior and well-being [13,38,41], leading us to the hypothesis that financial behavior, such as engaging in investment activities, is positively associated with overall financial well-being.

*Hypothesis 3: Financial behavior is strongly associated with subjective financial well-being.*

The positive impact of financial behavior on financial well-being is mediated by individual differences [42,43], emphasizing the need to incorporate psychological variables into a model of subjective financial well-being.

Individual differences refer to cognitive response styles that shape how individuals interpret contexts, process information, and make decisions [44]. This study focuses on impulsiveness, future orientation, and maximization, which are measured through validated scales to provide a score in those domains. Individual differences in cognitive response styles moderate how individuals emotionally and cognitively evaluate their economic situations [43], as they are closely tied to financial needs, expectations, goals, and ambitions [45,46].

Impulsiveness, characterized by actions taken without control, planning, or consideration of the consequences, is negatively associated with subjective financial well-being due to its link with lower self-control [47], poor quality decisions [48] and planning [49]. This often results in poor financial choices, but also compulsive buying and excessive debt over time.

Future orientation is defined as a time perspective focused on the future, by setting and achieving personal goals. Its positive association with goal-setting and long-term asset accumulation [34] positions future orientation as a potential predictor of subjective financial well-being [50].

Maximization describes people who constantly search for the best possible option [51] and set higher standards for themselves [52,53]. Individuals with high maximization tendencies are typically less satisfied with their life [51] and decisions [54,55] due to heightened sensitivity to the negative aspects of complex choices [56], suggesting a negative correlation with subjective financial well-being.

In terms of individual psychological differences and subjective financial well-being, impulsiveness has often been indirectly linked to low levels of financial well-being [48,57]. The indirect negative role of maximization warrants further investigation, particularly for the need to distinguish it from regret [51]. Future orientation has shown a fundamental role in subjective financial well-being, as it aligns with goal-setting theory [50].

*Hypothesis 4: Cognitive response styles are differently related to subjective financial well-being. Impulsiveness and maximization are expected to negatively correlate with subjective financial well-being, while future orientation is expected to correlate positively.*

Trust has been defined as the belief in the dependability of governmental institutions and financial professionals to deliver on their promises [58]. Trust in governmental institutions is tipically measured through surveys, assessing aspects such as equality, protection, trust, financial policy, and bureaucracy efficiency. Trust in financial professionals can be measured by evaluating the perceived benevolence of their financial offerings and the overall reputation of the financial institutions they represent. In this study, the scale measuring trust in financial

professionals contains items about the perceived benevolence of financial professionals as well as broader perceptions of the financial institutions where they work (see S1 Appendix in S1 File). Trust in governmental institutions and financial professionals has been included into our framework due to its significant influence on individual perception [16], privacy concerns [59,60], and perceived security of the economic situation, respectively at a general and individual level [61]. Trust has been demonstrated to be positively and robustly associated with financial well-being [62]. Trust is the foundation of all social systems [63] and a source of resilient responses in aversive situations [64]. Given that trust in financial professionals varies across European countries, it is important to analyze how these differences influence financial well-being [24].

Trust in the governmental institutions plays a central role in developing and maintaining social order and individual well-being [63]. Although prior research has not extensively examined the connection between trust in financial professionals and subjective financial well-being, this study addresses this gap by including trust in both financial professionals and governmental institutions in our model.

*Hypothesis 5: Trust in governmental institutions and financial professionals positively relates to subjective financial well-being.*

## Methodology

### Participants

Out of 393 participants who took part in the survey, the data of eight participants were not included in the final analysis because at the time they were not living in Germany or Italy. The final sample ($N = 385$) consisted mostly of undergraduate students ($n = 299$), with a relatively greater proportion of women ($n = 277$), with a mean age of 24.5 years ($SD = 4.19$, range 18–43 years). Half of the sample lived in Germany ($n = 192$), and the other half was in Italy ($n = 193$).

The participants represented a convenience sample with a high proportion of students who were within an age range of 18–29 years old, defined as "emerging adult" population [12]. We employed a convenience sample which is adequate to answer specific research questions about a specific population [65], such as the emerging adult population.

### Procedure

The study was conducted from 15/02/2018 to 17/02/2020. Participants were recruited by distributing the survey invitation via social media and university courses at the Università Cattolica del Sacro Cuore in Milan, Italy, and at the University of Applied Sciences Würzburg-Schweinfurt in Germany (see S2 File). Before starting the study, participants were informed that the survey was anonymous, self-reported, voluntary, and without any form of incentive, and that they were allowed to leave the study at any time. They were informed at the beginning of the online questionnaire that their information would have been treated as confidential, and used only for academic and scientific purposes. By sending the Google Form, they provided their informed consent for the data treatment and incomplete forms were not recorded or reported here. As this study was non-interventional and adhered to ethical guidelines, it received an ethics waiver from the University of Konstanz [RefNo: IRB24KN009–11/w].

The online questionnaire, designed as a Google Form, was originally written in both Italian and English, by using validated versions of the psychological scales (available in the Open Science Framework https://osf.io/hmqfe/, along with the full data set and the script used for the analyses – https://osf.io/hmqfe/). It was later translated into German with the support of

two independent translators. The survey took about 30 minutes to complete and included informed consent and sections on socio-demographic information, financial habits, knowledge about the pension system, financial literacy, financial behavior, individual psychological differences, and trust. Following, we describe the measures that were used for this study.

## Measures

Standardized scales for all measures were administered to participants, except for those items related to trust in financial professionals, that were created *ad hoc* (see S1 Appendix in S1 File).

The *Multidimensional Subjective Financial Well-Being Scale* for emerging adults [9] includes a four-point Likert scale (1 = Almost never true; 4 = Almost always true) to evaluate ten statements about their individual financial situation and other people's perception of their resources. It consists of five factors computed out of 25 items: "General Subjective Financial Well-being", "Financial Future", "Money Management", "Having Money", and "Peer comparison". Subjective financial well-being was measured with the "General Subjective Financial Well-being" factor of the scale, this factor is suggested to be used alone when the interest of the research is on subjective financial well-being at a general level [9]. The index score is computed from the mean of the items. The "General Subjective Financial Well-being" factor consisted of 10 questions about people's perception of their economic condition (e.g., "I am satisfied with my present financial situation").

## Financial competencies

The *Financial Literacy Questionnaire* [66,67] and the *Eight Knowledge Questions* [34] are, respectively, three-item and eight-item tasks measuring financial literacy. The index scores are computed for both measures by summing the values assigned to individual items. Altogether, they tested the participants' understanding of financial concepts, such as interest, inflation, stock market, time value of money, risk and return, and diversification. Examples of items are "Suppose you had €100 in a savings account and the interest rate was 2% per year. After five years, how much do you think you would have in the account if you left the money to grow?" [66,67] or "Imagine that five brothers are given a gift of 1000 €. If the brothers have to share the money equally how much does each one get?" [34] (currency adapted to EUR).

The *Financial Behavior Score* [34] consists of a total of nine questions about specific situations such as "considered purchase", "timely bill payment", "attention to financial affairs", "long-term financial goal setting", "responsibility of a household budget", "active saving", "choice of financial products" and the practice of "borrowing to make ends meet". The answers are given using a five-point Likert scale (1 = Never; 5 = Always). The index score is obtained from the sum of the values assigned to the individual items. A sample item is "I keep a close personal watch on my financial affairs".

## Cognitive response styles

The *Barratt Impulsiveness Scale (BIS-11)* measures motor, cognitive, and non-planning impulsiveness [68]. The 30-item scale (e.g., "I do things without thinking") describes daily situations, which should be evaluated on a four-point Likert scale (1 = Never/rarely; 4 = Almost always/always), and the index score is obtained by summing the three main subscales (motor, cognitive, and non-planning impulsiveness).

The *Future Orientation Scale* is a survey that describes the ability to manage financial affairs from a long-term strategic perspective ("Attitudes Toward the Longer Term" in Atkinson & Messy [34]). The three items are evaluated on a five-point Likert scale (1 = I totally agree; 5 = I entirely disagree). The index score is calculated from the mean of the items. An example item is "I find it more satisfying to spend money than to save it for the long term" [34].

The *Maximization and Regret Scale* [51] is a seven-point Likert scale (1 = I completely disagree; 7 = I completely agree) that consists of 18 items (e.g., "I often find myself fantasizing about how my life could be different compared to what it is like today). The index score is obtained from the mean values of all the items. The answers are aggregated into four factors: Regret, openness, shopping behavior, and high standards. The three factors, except regret, constitute the maximization score.

### Trust in governmental institutions and financial professionals

The *Assessment Criteria for Social Sustainability* (Sustainable Governance Indicators, 2016) [69,70] is a qualitative measure adapted to this study to measure trust in governmental institutions. It is a ten-point Likert scale (1 = Not at all; 10 = Very much) with five questions about equality, perceived protection, trust, financial policy, and bureaucracy efficiency (e.g., "I feel protected by the policy of the country where I live"). The index score is calculated by summing the values assigned to the items.

The *Trust in Financial Professionals Scale* was designed for this study to investigate primarily the perception of financial professionals but also of the financial institutions to which they belong. It is a five-point Likert scale (1 = It does not describe me at all; 5 = It totally describes me), constituted by ten items analyzing the extent of subjective trust in financial professionals (e.g., "Trust in financial products offered by professionals", "Trust in financial professionals showing empathy", etc. – see S1 Appendix in S1 File). The index score is obtained from the mean of the items. An example item is "I trust when a banking professional proposes me a financial product".

### Analysis

Multiple regression analysis was used to test whether financial well-being is predicted by socio-demographic variables, financial competencies, cognitive response styles, and trust. To assess the effects of different measures on financial well-being, we built two multiple linear regression models in R [71]. The multiple linear regression analysis used the following equation:

$$Y = a + b_1 X_1 + b_2 X_2 + \ldots + b_n X_n,$$

where $Y$ represents the subjective financial well-being score, $X_1$ through $X_n$ represent independent variables, $a$ is a constant term, and $b_1$ through $b_n$ are the regression coefficients for each independent variable.

We tested for multicollinearity by calculating the variance inflation factor (VIF) and checked the normality of the residuals with the Kolmogorov-Smirnov test. All reported effects are correlative, so they do not imply causality; rather, we interpreted the models in terms of variables that explain the variance in subjective financial well-being. Variables were centered around the group mean to increase the interpretability of the results. In addition, we calculated the reliability of the survey scales.

### Results

We calculated the descriptive statistics and reliability of the survey scales (see Table 1). The internal consistency of the survey scales was at a conventionally acceptable level ($\alpha \geq 0.70$), except for the financial behavior scale ($\alpha = 0.54$). Importantly, it was very good for subjective financial well-being ($\alpha \geq 0.9$).

We conducted a multiple regression analysis to test the study hypotheses. Model 1 represents the theoretical account, in which socio-demographic variables, financial literacy, financial

**Table 1. Descriptive statistics and reliability of the survey scales.**

| Scale | M | SD | Range | Cronbach's $\alpha$ |
|---|---|---|---|---|
| Subjective financial well-being | 2.73 | 0.69 | 1–4 | .92 |
| Financial literacy | 0.63 | 0.23 | 0.07–1 | .60 |
| Financial behavior | 6.32 | 2.02 | 1–11 | .54 |
| Impulsiveness | 13.54 | 1.93 | 8.43–19.75 | .77 |
| Future orientation | 3.31 | 0.80 | 1–5 | .70 |
| Trust in financial professionals | 3.11 | 0.58 | 1–4.5 | .70 |
| Trust in governmental institutions | 21.72 | 9.64 | 5–50 | .92 |
| Maximization | 3.95 | 0.85 | 1.78–6.17 | .82 |

behavior, future orientation, and impulsiveness are related to subjective financial well-being. The results of the regression indicated that the predictors explained 7% of the variance ($R^2 = 0.07$, adj. $R^2 = 0.05$, $F(9, 371) = 3.06$, $p = .0015$). Model 2 includes all variables of Model 1, but it incorporates also maximization, trust in financial professionals, and trust in governmental institutions. The results of the regression indicated that the predictors explained 11% of the variance ($R^2 = 0.11$, adj. $R^2 = 0.08$, $F(12,368) = 3.61$, $p < .001$). In both models, the normality of the residuals was confirmed by the Kolmogorov-Smirnov test ($p$s > 0.05). The variance inflation factors for each predictor were less than 2 in both models, suggesting that multicollinearity was not an issue in the models. The addition of further maximization and trust variables to Model 2 improved the model fit compared to Model 1, $F(3,371) = 6.53$, $p = .002$, and did not substantially change the effect sizes of the other variables (see Table 2 and Fig 2), so we report

**Table 2. Regression coefficients for financial well-being.**

| Variable | Model 1 | | | Model 2 | | |
|---|---|---|---|---|---|---|
| | B | SE | p | B | SE | p |
| (Intercept) | 2.66 | 0.05 | **<0.001** | 2.74 | 0.06 | **<0.001** |
| Age | −0.03 | 0.01 | **0.009** | −0.02 | 0.01 | **0.024** |
| Country of residence/ Germany | 0.13 | 0.07 | 0.08 | −0.03 | 0.09 | 0.76 |
| Job/ Manager, professional, services | 0.07 | 0.12 | 0.56 | 0.05 | 0.12 | 0.67 |
| Job/ Agriculture, craftsman, operator, handwork | 0.04 | 0.14 | 0.78 | 0.03 | 0.13 | 0.84 |
| Job/ Unemployed | −0.43 | 0.26 | 0.10 | −0.49 | 0.26 | 0.06 |
| Financial literacy | 0.11 | 0.15 | 0.48 | 0.08 | 0.15 | 0.59 |
| Financial behavior | 0.02 | 0.02 | 0.28 | 0.02 | 0.02 | 0.24 |
| Impulsiveness | −0.06 | 0.02 | **0.003** | −0.04 | 0.02 | **0.046** |
| Future orientation | −0.11 | 0.04 | **0.011** | −0.11 | 0.04 | **0.011** |
| Trust in financial professionals | | | | 0.12 | 0.06 | 0.051 |
| Trust in governmental institutions | | | | 0.01 | 0.00 | **0.023** |
| Maximization | | | | −0.07 | 0.04 | 0.077 |
| $R^2$ | 0.07 | | | 0.11 | | |
| $R^2$ adjusted | 0.05 | | | 0.08 | | |

*Note.* N = 381. Unstandardized standard errors are presented. We examined the impact of socio-demographic characteristics and individual differences in financial competencies, cognitive response styles, and trust on financial well-being ratings. In Model 1, we entered socio-demographic variables and variables from the theoretical model of well-being such as financial literacy, impulsiveness, financial behavior, and future orientation [13] to predict financial well-being. In Model 2, trust and maximization were additionally used as predictors.

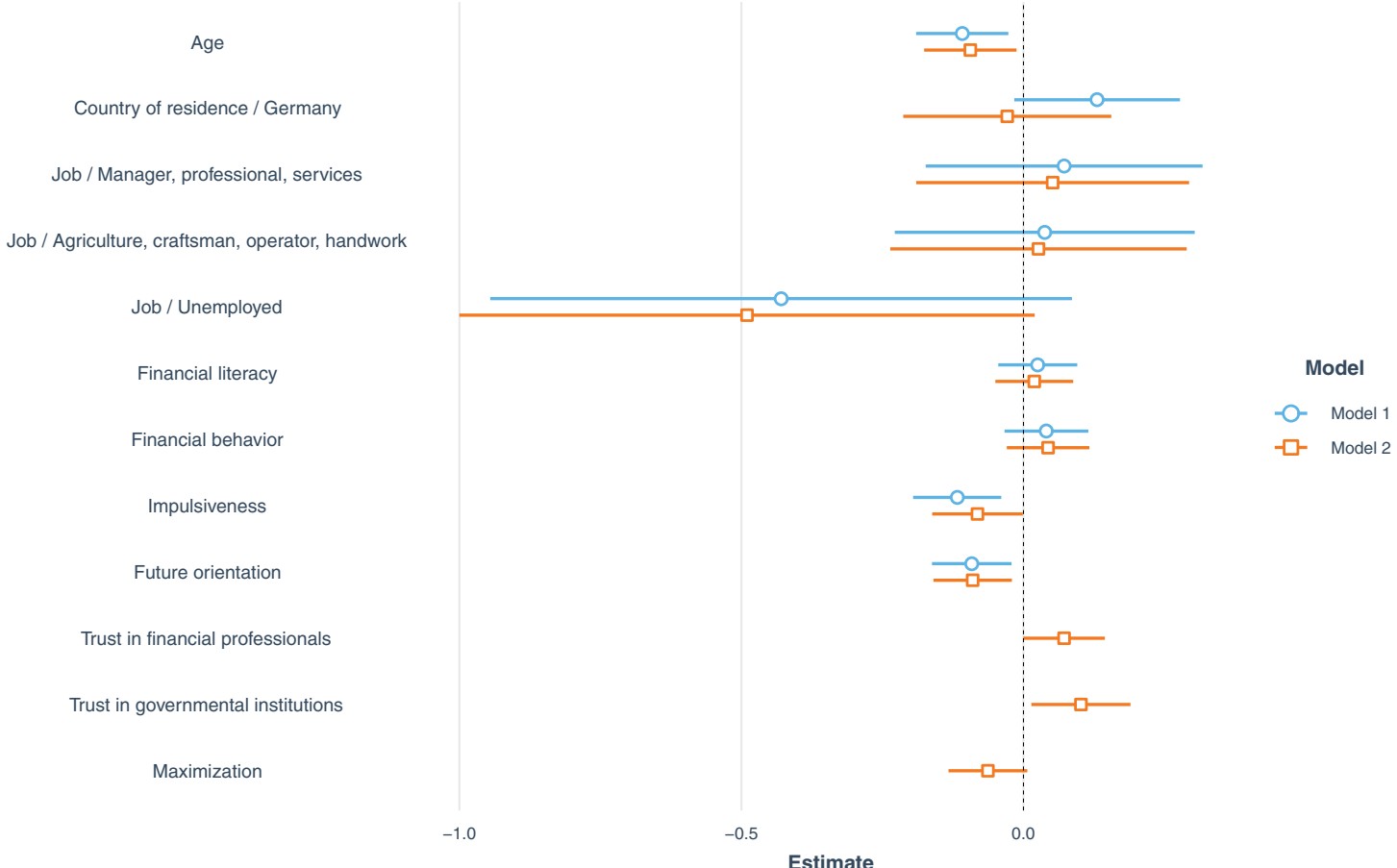

**Fig 2. Standardized regression coefficients for financial well-being.** *Note.* $N$ = 381. 95% confidence intervals are calculated based on standardized standard errors.

the estimates of Model 2. The intercept estimate ($b$ = 2.74, $SE$ = 0.06) represents the financial well-being score of an Italian student whose age and scores on all measured variables are at the average level.

Increase in age was negatively related to subjective financial well-being, $b$ = - 0.02, $SE$ = 0.01, $p$ = 0.024. Impulsiveness and future orientation were both negatively related to well-being, $b$ = - 0.04, $SE$ = 0.02, $p$ = 0.046, and $b$ = - 0.11, $SE$ = 0.04, $p$ = 0.011, respectively. Trust in governmental institutions was positively associated with financial well-being – the more participants trusted governmental institutions, the higher their subjective financial well-being, $b$ = 0.01, $SE$ = 0.005, $p$ = 0.023. There were no statistically significant differences in reported financial well-being between Germany and Italy, $b$ = - 0.03, $SE$ = 0.09, $p$ = 0.76.

## Exploratory analysis

The measure of financial literacy was observed to be positively correlated with the measure of financial behavior, $r$ = 0.15, $p$ = 0.003.

As the financial behavior scale showed a low level of internal consistency ($\alpha$ = 0.54), nine questions of the scale were analyzed using a principal component factor analysis with varimax rotation (orthogonal). The analysis revealed three factors from the financial behavior score [34] that explained a total of 34% of the variance. Factor 1 "Caring for financial matters"

comprised 4 items (Q1, Q2, Q3, Q4) that explained 13% of the variance with factor loadings from .38 to.61. Factor 2 "Active saving" contained 1 item (Q7), which explained 12% of the variance with the factor loading of 1.00, and factor 3 "Responsibility" was comprised of 2 items (Q5, Q6) that explained 9% of the variance with factor loadings from .54 to.63.

The three scores ("Caring for financial matters", "Active saving", and "Responsibility") were separately included in a multiple regression instead of financial behavior in Model 2. The caring for financial matters score was positively related to subjective financial well-being ($b = 0.32$, $SE = 0.13$, $p = 0.017$), whereas the effects of correlations with the other scores were not significant.

## Discussion

The current research explores factors that are related to subjective financial well-being in the emerging adult population by considering socio-demographic, psychological and attitudinal components. We drew attention to the necessity of providing an integrated overview of the factors involved in subjective financial well-being, with a specific focus on the emerging adult population in Europe. After testing the reliability of the survey scales, we compared two models of subjective financial well-being.

The comparison between Model 1 [13] and Model 2 highlighted the added value of a model that emphasizes psychological variables and trust. While individual differences are established as central to subjective financial well-being [18], our study extends this by examining how psychological factors shape individuals' interpretation of their context and influence the parameters that are involved in the decision-making processes [45,46]. By incorporating two distinct measures of trust — trust in governmental institutions and trust in financial professionals — our study adds a critical layer of attitudinal characteristics that link the individual to the broader societal context. This approach allows for a more nuanced understanding of how trust, as an attitudinal component toward society, is associated with subjective financial well-being.

### Contributing factors to the subjective financial well-being model

In line with Kempson and colleagues' theoretical framework [13], we observed a significant contribution of age and specific cognitive response styles to subjective financial well-being. Younger age was related to higher well-being, whereas higher impulsiveness and future orientation were associated with lower subjective financial well-being. In addition, the "Caring for financial matters" score (a sub-component of financial behavior) and trust in governmental institutions were positively related to subjective financial well-being.

With regard to the relationship between socio-demographic variables and subjective financial well-being (our first research question), our findings on age align with prior research showing that older adults face more financial concerns. A study by Riitsalu and van Raaij [72] showed that older people in Italy and Germany experience respectively higher stress for money management and less future financial security. Similarly, studies from the United Kingdom [23] indicate that older age reduces the positive effect of general health on subjective financial well-being, further supporting the positive relationship between younger age and subjective financial well-being. This highlights the direct connection between age and subjective financial well-being where age is considered a socio-demographic indicator from a comprehensive perspective.

For our second research question, we observed that caring for financial matters (a sub-component of financial behavior) and some of the measured individual differences (impulsiveness, future orientation) were significantly related to subjective financial well-being. With regard to financial competencies, differently from previous suggested literature [34,73] suggesting that financial behavior is homogenous, our exploratory analysis grouped financial

behavior into three factors: caring for financial matters, active saving, and taking financial responsibility. The caring for financial matters factor has shown a positive relationship with financial well-being, consistent with findings from studies in Japan, the United States, and the Netherlands, which indicated that engaging in financial activities may enhance self-confidence [38,40] and thus financial well-being [74]. This suggests that the financial behavior measure might benefit from a further focus on people's caring for financial matters to better understand the positive relationship with subjective financial well-being. In line with prior research, which emphasized the importance of emerging adult's engagement in financial matters as a financial well-being indicator, our study further supports the connection between financial engagement and subjective financial well-being in emerging adulthood.

Regarding individual differences in cognitive response styles, significant relationships with subjective financial well-being were observed for impulsiveness and future orientation. Although, the latter showed an opposite direction from what was hypothesized.

The negative relationship between impulsiveness and subjective financial well-being aligns with previous research, which links impulsiveness, characterized by a lack of planning, to lower levels of perceived financial security and freedom of choice — key components of subjective financial well-being [48,75]. Previous studies conducted in Sweden showed that poor self-control is negatively related to subjective financial well-being [47], and that impulsive consumption is negatively related to objective financial well-being [48]. A meta-analysis highlighted that indebtedness is a sign of low financial well-being, particularly in young populations in the United States [49]. Our findings advance the research by demonstrating that impulsiveness as a personality trait, beyond just behavior, negatively impacts subjective financial well-being, beyond the general financial well-being.

Contrary to our hypothesis, future orientation was found to be negatively related to subjective financial well-being. This may be explained by the fact that individuals who are more future-focused tend to worry about the future and forego short-term gratification [76,77]. This finding advances our knowledge about how future orientation relates to subjective financial well-being. Previous studies across OECD countries have shown differences among countries, highlighting that people living in Albania, the Czech Republic, Germany, Hungary, Peru, and the British Virgin Islands are characterized by high future orientation [34]. While a focus on the future supports virtuous financial behavior and positively impacts objective financial well-being [50], it does not necessarily enhance subjective financial well-being, particularly among emerging adults in Italy and Germany, who may experience higher anxiety and uncertainty about the future.

Lastly, our study contributes to the growing evidence supporting the role of trust in subjective financial well-being by highlighting the positive relationship with trust in governmental institutions. This is consistent with recent findings suggesting that when governments are perceived as handling financial challenges effectively, people report higher financial and general well-being, as seen in Sweden [62]. High levels of trust may also encourage greater risk-taking, especially in relation to intangible financial products like investments and life insurance [78,79]. A study conducted in Colombia also demonstrated the importance of trust in shaping perceptions of economic situation at both general and individual levels [61].

Our findings lend further evidence to the notion that subjective financial well-being is shaped by a complex interplay of socio-demographic characteristics, individual differences, and attitudinal factors that link individuals to society.

## Limitations and future research

To increase the interpretability of our findings, it is fundamental to discuss the limitations of our study and lay the groundwork for future research.

First, we employed a convenience sample because it is considered acceptable for addressing specific research questions about specific populations [65]. However, this approach limits representativeness and generalizability to the broader population of emerging adults. Future research should employ probabilistic sampling methodologies to enhance generalizability. For instance, online platforms (e.g., Prolific) enable pre-screening of participants, which facilitates the study of specific populations while improving generalizability.

Secondly, for simplicity and practicality, we focused on only two countries, each representing a model of transition to adulthood [4]. Examining only one country per model restricts the generalization of results to other countries within those models. Including additional countries could clarify the role of different models of transition to adulthood in subjective financial well-being. Future research could adopt a large-scale study approach involving collaborations with laboratories across Europe for translation and data collection [80].

Thirdly, we used traditional measurements of financial literacy, financial behavior, and future orientation in the financial domain. While these measures were appropriate, incorporating additional ones could provide a more nuanced understanding of their relationship with subjective financial well-being.

The relationship between financial literacy, financial behavior, and subjective financial well-being remains complex. Although a weak positive correlation between financial literacy and financial behavior was observed, neither significantly impacted subjective financial well-being. Future research should identify which components of financial literacy and financial behavior contribute more effectively to subjective financial well-being. For financial literacy, employing measures of perceived financial knowledge, along with the measure of objective financial literacy [28,29] could clarify whether one of those facets of financial literacy explains subjective financial well-being. For financial behavior, our exploratory analysis suggests that this measure may comprise multiple components, with "Caring for financial matters" showing a positive relationship with subjective financial well-being. Future research should examine whether the components observed here for the Financial Behavior Score [34] scale are replicable across different populations, and further investigate the role of "Caring for financial matters" in subjective financial well-being.

Future orientation is crucial in financial behavior and decision-making, yet the relationship between future orientation and subjective financial well-being remains unclear. Future research should explore why we observed that individuals with high future orientation experience lower subjective financial well-being. In this study, we used a measure commonly employed in financial well-being research. However, future studies could include scales examining the influence of the future on current decisions and the perceived temporal distance of the future, such as the Future Time Orientation Scale by Coscioni and colleagues [81].

Finally, we developed an *ad hoc* measure of trust in financial professionals, which demonstrated adequate reliability. However, the lack of a validated scale for this measure represents a limitation of our study. Future research should validate psychological scales measuring trust in financial professionals and clarify its role in subjective financial well-being. Such scales should also be built from response options recently developed in survey methodology following more widespread use of online surveying [82], visual analogue scales (VAS). These scales have been shown to provide greater precision and usability, lead to less psychological reactance among participants, and provide more adequate characteristics for parametric statistical analyses [83,84].

Despite these limitations and the need for future research, the present study offers valuable insights with important theoretical and practical implications.

## Theoretical and practical implications

People face a variety of financial challenges throughout their lives. While the current study focused primarily on the emerging adults in two European countries representing respectively the Northern and the Mediterranean models of emerging adulthood, examining subjective financial well-being in other countries and among older populations, especially in light of global aging and unstable pension systems, would provide valuable new insights.

The promising role of trust in financial professionals, coupled with the general acceptance of financial professionals among emerging adults [85], highlights the potential for designing interventions that foster trust to improve subjective financial well-being.

Focusing on subjective financial well-being can contribute to the development of a comprehensive model of well-being. Incorporating individual differences in cognitive response styles and trust can inform the design of targeted interventions to improve well-being, given the important role that subjective financial well-being plays in the psychological and overall well-being of the emerging adult population [3].

## Conclusions

Once the assumption that two emerging adults with similar material resources will have the same level of financial well-being has been overcome, it becomes important to understand the factors that are related to financial well-being. Drawing on the definition of financial well-being as multidimensional, our study aimed to identify the specific factors contributing to subjective financial well-being from a comprehensive perspective. Our findings suggest that socio-demographic, individual differences in cognitive response styles and attitudinal characteristics shape subjective financial well-being in emerging adulthood. Specifically, we found that emerging adults with high levels of subjective financial well-being tend to be younger, more concerned with financial matters (a subcomponent of financial behavior), less impulsive, and have greater trust in governmental institutions. However, our findings provide only partial confirmation of the hypothesized factors, as no significant relationships were observed between subjective financial well-being and variables such as country of residence, job type, financial literacy, financial behavior (classic measure), maximization, and trust in financial professionals.

While this study provides valuable insights, its reliance on a convenience sample and the focus on only two countries to represent models of transition to adulthood [4] constitute limitations in our study. Future research should employ probabilistic sampling methodologies and include participants from additional countries to test and improve the generalizability of these findings.

This research lays the groundwork for a model of subjective financial well-being in emerging adults that integrates socio-economic, psychological, and attitudinal characteristics. Evaluating the psychological dimensions underlying financial well-being is crucial for both individuals and institutions. Equipping individuals with psychological resources helps them navigate complex financial challenges, while institutions benefit from evidence-based guidelines designed to enhance subjective financial well-being. By identifying the factors related to subjective financial well-being within a comprehensive model, this study contributes to the empirical foundation for evidence-based guidelines and provides valuable insights for future research and practical applications.

## Supporting Information

**S1 File. Trust in Financial Professionals scale.**
This file contains the *Trust in Financial Professionals Scale*, developed specifically for this study to assess individuals' trust in financial professionals.

**S2 File. Inclusivity in Global research.**
This file contains additional information regarding the ethical, cultural, and scientific considerations specific to inclusivity in global research.

## Acknowledgments

The authors would like to express their gratitude to all the people and participants who have made this project possible, along with the external translators, Ela Scarì and Federica Carrara, who greatly helped in translating the survey into German.

## Author contributions

**Conceptualization:** Maria Rosa Miccoli, Paola Iannello.

**Data curation:** Maria Rosa Miccoli, Yury Shevchenko.

**Formal analysis:** Yury Shevchenko.

**Investigation:** Maria Rosa Miccoli.

**Methodology:** Maria Rosa Miccoli, Yury Shevchenko, Paola Iannello.

**Supervision:** Paola Iannello, Ulf-Dietrich Reips.

**Visualization:** Yury Shevchenko.

**Writing – original draft:** Maria Rosa Miccoli.

**Writing – review & editing:** Yury Shevchenko, Paola Iannello, Ulf-Dietrich Reips.

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
