## [Decision Letter · Decision Letter 0]

23 Dec 2024

PONE-D-24-41072Factors Shaping Subjective Financial Well-being in Emerging Adults: A Comparative Study of Italy and GermanyPLOS ONE

Dear Dr. Miccoli,

Thank you for submitting your manuscript to PLOS ONE. After careful consideration, we feel that it has merit but does not fully meet PLOS ONE’s publication criteria as it currently stands. Therefore, we invite you to submit a revised version of the manuscript that addresses the points raised during the review process.

The authors must take into consideration all the reviewer's suggestions in order to publish this article.

We look forward to receiving your revised manuscript.

Kind regards,

José Manuel Santos Jaén

Academic Editor

PLOS ONE

Additional Editor Comments :

The authors must take into consideration all the reviewer's suggestions in order to publish this article.

Reviewers' comments:

Reviewer's Responses to Questions

**Comments to the Author**

1. Is the manuscript technically sound, and do the data support the conclusions?

Reviewer #1: Partly

2. Has the statistical analysis been performed appropriately and rigorously? 

Reviewer #1: No

3. Have the authors made all data underlying the findings in their manuscript fully available?

Reviewer #1: No

4. Is the manuscript presented in an intelligible fashion and written in standard English?

Reviewer #1: Yes

5. Review Comments to the Author

Reviewer #1: In the Abstract, the expression "F(3, 371) = 6.53, p = .002" is unnecessary and can be omitted.

In the section Theoretical Analysis and Hypothesis Development, adding a visual representation or diagram would help readers understand the concepts more effectively.

In the Analysis section, when introducing multiple regression analysis, it would be beneficial to provide the exact regression equation used in the study.

The Conclusion section should include a discussion of the study’s limitations.

In the Conclusion section, the suggestions for “further research” should be more specific and detailed.

It would be better to switch the order of the Theoretical and Practical Implications section and the Conclusions section for improved flow and readability.

6. PLOS authors have the option to publish the peer review history of their article (what does this mean? ). If published, this will include your full peer review and any attached files.

**Do you want your identity to be public for this peer review?** For information about this choice, including consent withdrawal, please see our Privacy Policy .

Reviewer #1: No

---

## [Author Response · Author response to Decision Letter 1]

5 Feb 2025

- Additional Editor Comments :The authors must take into consideration all the reviewer's suggestions in order to publish this article.

As required, we carefully took into consideration all the reviewer's suggestions and updated the manuscript accordingly. Thank you for giving us this opportunity.

Answers to the reviewer:

- In the Abstract, the expression "F(3, 371) = 6.53, p = .002" is unnecessary and can be omitted.

Done.

- In the section Theoretical Analysis and Hypothesis Development, adding a visual representation or diagram would help readers understand the concepts more effectively.

On page 8, we have now included a visual representation [Fig 1] of the factors hypothesized to be related to subjective financial well-being. The figure has been uploaded to the submission portal.

- In the Analysis section, when introducing multiple regression analysis, it would be beneficial to provide the exact regression equation used in the study.

Done [p. 16]. We constructed two multiple linear regression models. Model 1 includes variables identified in the framework proposed by Kempson and colleagues. Model 2 builds upon Model 1 by incorporating additional variables, specifically trust, and maximization, to examine their effects on subjective financial well-being. We tested the fundamental assumptions of multiple regression. Reliability measures for the survey scales were included to demonstrate the soundness of the instruments used.

The data and scripts used for the analyses are publicly available on the Open Science Framework project (https://osf.io/hmqfe/), and this has now been clarified in the manuscript.

- The Conclusion section should include a discussion of the study’s limitations.

Thank you for your valuable suggestion. We now included a “Limitations and future research” paragraph [p. 23-25] in the discussion section, where we specify the limitations of our study to facilitate the interpretation of our results. We have also briefly referenced the key limitations in the Conclusion [p. 27] to further align with your suggestion.

- In the Conclusion section, the suggestions for “further research” should be more specific and detailed.

Done. We further increased the specificity of the suggestions for future research, and we moved such suggestions from the “Theoretical and Practical implications” to the new paragraph “Limitations and future research” [p. 23-25].

- It would be better to switch the order of the Theoretical and Practical Implications section and the Conclusions section for improved flow and readability.

Thank you for your suggestion. We have switched the order of the sections to improve flow and readability, as recommended. In alignment with guidelines for scientific writing, the “Theoretical and Practical Implications” section has now been integrated with the Discussion, where it logically complements the interpretation of findings [p.25-26]. Additionally, we have further refined the Conclusions section to summarize the study's key contributions and takeaways.

---

## [Decision Letter · Decision Letter 1]

25 Feb 2025

Factors Shaping Subjective Financial Well-being in Emerging Adults: A Comparative Study of Italy and Germany

PONE-D-24-41072R1

Dear Dr. Miccoli,

We’re pleased to inform you that your manuscript has been judged scientifically suitable for publication and will be formally accepted for publication once it meets all outstanding technical requirements.

Kind regards,

José Manuel Santos Jaén

Academic Editor

PLOS ONE

Additional Editor Comments (optional):

Based on the reviewer's examination, I consider that the article can be published. Congratulations to the authors for their work.

Reviewers' comments:

Reviewer's Responses to Questions

**Comments to the Author**

1. If the authors have adequately addressed your comments raised in a previous round of review and you feel that this manuscript is now acceptable for publication, you may indicate that here to bypass the “Comments to the Author” section, enter your conflict of interest statement in the “Confidential to Editor” section, and submit your "Accept" recommendation.

Reviewer #1: All comments have been addressed

2. Is the manuscript technically sound, and do the data support the conclusions?

Reviewer #1: Yes

3. Has the statistical analysis been performed appropriately and rigorously? 

Reviewer #1: Yes

4. Have the authors made all data underlying the findings in their manuscript fully available?

Reviewer #1: Yes

5. Is the manuscript presented in an intelligible fashion and written in standard English?

Reviewer #1: Yes

6. Review Comments to the Author

Reviewer #1: The authors have appropriately modified the manuscript according to the reviewer's comments. The revised manuscript is now satisfactory. I would like to thank the authors for their efforts.

7. PLOS authors have the option to publish the peer review history of their article (what does this mean? ). If published, this will include your full peer review and any attached files.

**Do you want your identity to be public for this peer review?** For information about this choice, including consent withdrawal, please see our Privacy Policy .

Reviewer #1: No

---

## [Editor Report · Acceptance letter]

PONE-D-24-41072R1

PLOS ONE

Dear Dr. Miccoli,

I'm pleased to inform you that your manuscript has been deemed suitable for publication in PLOS ONE. Congratulations! Your manuscript is now being handed over to our production team.

Kind regards,

on behalf of

Dr José Manuel Santos Jaén

Academic Editor

PLOS ONE